# Influence of Sintering Process on Microstructure and Mechanical Properties of Ti(C,N)-Based Cermet

**DOI:** 10.3390/ma13183938

**Published:** 2020-09-05

**Authors:** Kaixun Ji, Yanxin Meng, Fuzeng Wang, Yousheng Li

**Affiliations:** 1Institute of Manufacturing Engineering, Huaqiao University, Xiamen 361021, China; jikaixun95@163.com (K.J.); tanggao1208@163.com (Y.M.); 2Institute of Manufacturing Engineering, Fujian Engineering Research Center of Intelligent Manufacturing for Brittle Materials, Xiamen 361021, China; 3Xiamen Golden Egret Special Alloy Co., Ltd., Xiamen 361006, China; li.yousheng@cxtc.com

**Keywords:** Ti(C,N)-based cermet, vacuum sintering, hardness, fracture toughness

## Abstract

In this study, a Ti(C,N)-based cermet material was prepared through vacuum sintering. The research also investigates how holding time and maximum sintering temperature influence the material microstructure and mechanical properties. X-ray diffraction (XRD), energy dispersive spectroscopy (EDS) were used to analyze the composition of the cermet. The microstructure of the cermet was analyzed and examined using a scanning electron microscope (SEM). A Vickers hardness tester was used to test the mechanical properties of the materials. As indicated by testing results, the hardness of the material decreases as the temperature of sintering increases, and its fracture toughness increases gradually as holding time increases. Ti(C,N)-based cermet manifested the optimal mechanical properties when sintering was conducted under 1400 °C with 80 min of holding time. Moreover, the material microstructure is significantly affected by the sintering process. The grain size of Ti(C,N) cermets increases as the sintering temperature increases. The microstructure tends to be uniform and the complete core-rim structures are established as the holding time increases.

## 1. Introduction

Ti(C,N)-based cermets are already widely applied as die materials, parts resistant to wear and cutting instruments in view of good thermal performance, chemical stability and high hardness [1,2]. Typically, Ti(C,N)-based cermets comprise a Co/Ni binder phase and a Ti(C,N)-based phase. Ti(C,N)-based cermets already manifest great potential in taking the place of common tungsten carbide as their hardness is high under low coefficients of friction and high temperatures [3,4].

At present, as technologies of advanced sintering develop, diversified sintering methods concerning cermets have been developed, including vacuum sintering [5,6] and gas pressure sintering [7]. With the emergence of advanced sintering equipment, more and more studies have been conducted on sintering methods such as microwave sintering [8], hot-pressing sintering [9,10], high-pressure temperature sintering [11] and spark plasma sintering (SPS) [12]. These new methods of sintering provide new ideas for fabrication of cermet materials and boost development of cermet materials. In the industry, vacuum sintering is still the most commonly used sintering method in view of extensive features and high controllability [13,14].

As the toughness and strength of Ti(C,N)-based cermets are generally inferior to WC-Co cemented carbide, a lot of contributions are already conducted for enhancing toughness and strength of Ti(C,N)-based cermets through addition of different rare earth elements and alloying elements [15,16,17,18,19,20,21,22]. You et al. [15] and Dong et al. [16] found that W elements added mainly existed in an edge microstructure phase, so wettability between the Ti(C,N) hard phases and the bonding phase was improved. As reported by Liu et al. [17], Mo elements can also enhance wettability of the Ti(C,N) hard phases. Wu et al. [18] and Wang et al. [19] have demonstrated that TaC and NbC can improve resistance to high-temperature deformation and thermal shock resistance of Ti(C,N)-based cermets. As recently found by Lin et al. [20] and Wang et al. [21], through adding of VC, grain growth in Ti(C,N)-based cermets could be inhibited as Ti atom solubility was weakened under the liquid binder phase. Kang et al. [22] argued that ZrC can improve Ti(C,N)-based cermet resistance to oxidation under high temperatures. At present, most research on Ti(C,N)-based cermets highlight how diversified additives influence the material microstructure and mechanical properties. The sintering process is an important part of the manufacture of Ti(C,N)-based cermets [23], as it greatly affects its microstructure and mechanical properties.

In this paper, vacuum sintering was conducted to prepare Ti(C,N)-based cermet materials. To compare how this process influences cermet material microstructure and mechanical properties, three different temperatures of sintering and holding time were selected respectively in the experiment. In this way, analysis of how sintering temperature and holding time influenced cermet mechanical properties and microstructure was conducted.

## 2. Experimental Details and Processes

### 2.1. Preparation of Materials

In this experiment, raw materials Ti(C,N) powders contain Co, Ni, MoC (6.03 wt.% carbon), WC (5.91 wt.% carbon) and Ti(C_0.5_,N_0.5_) (13.25 wt.% carbon). Material powder parameters are shown in Table 1.

Figure 1 displays the morphology of Ti(C,N) powder.

All types of powder were milled in absolute ethyl alcohol by a high-energy planetary machine of ball milling (UBE-V0.4L, Hunan, China) for 12 h at the rate of 160 rpm. The ball-to-powder ratio was 10:1. After that, the powders were dried in a vacuum environment, blended with 3 wt.% of paraffin and compressed at 150 MPa. The molding blocks were placed in a vacuum sintering furnace (1800 °C/VVPgr-100-2000, Shanghai, China). The sintering chamber and heating elements of the vacuum sintering furnace was graphite tube. This paper used five different Ti(C,N)-based cermets sintering processes for comparison. The materials were sintered at a series of temperatures, 1400 °C, 1425 °C and 1450 °C respectively, with holding time lasting for 60 min. The other two sintering processes’ holding times were 40 min and 80 min when the sintering temperature was 1400 °C.

Figure 2 displays a flow chart of vacuum sintering of sintered Ti(C,N)-based cermets in this study.

The whole sintering process was carried out in the vacuum of 1 × 10^−2^ Pa. The dewaxing stage was realized under 600 °C. In the dewaxing stage, the heating rate was set at 5 °C/min. When the sintering temperature reached 1200 °C, the Ti(C,N)-based cermet entered the solid phase sintering stage. The heating rate was set 2.5 °C/min during sintering at the solid-phase stage, as a higher value would retard the core-rim structure formation, and a lower value would lead to inhomogeneous microstructure and grain coarsening [24].

### 2.2. Test of Mechanical Properties

The specimen was indentation tested with a Vicker hardness tester (DHV-1000, Shanghai, China). The applied force was 30 kg and the holding time was 20 s. The indentation crack length was measured by an optical microscope (KH-8700, Shanghai, China), as shown in Figure 3. During the test, each sample was measured five times then the arithmetic average taken for a more accurate comparison.

In this paper, the formula for calculating fracture toughness was proposed by Shetty et al. [25], Equation (1):(1)$$K_{IC}=A\sqrt{\mathrm{HV}}\sqrt{\frac{P}{\sum l}}$$
where HV is the hardness (N·m^−1^), *P* is the load (N), ∑l is the sum of crack lengths (mm) and *A* is a constant. $A=1/\left[3\left(1-V^{2}\right){\left(2^{\frac{1}{2}}\pi^{\frac{5}{2}}tan\theta \right)}^{\frac{1}{3}}\right],$ where V is Poisson’s ratio of the material and θ is the angle of the opposite faces of Vickers indenter. Its value is 2θ = 136° in our test. For *v* = 0.22 (typical of WC alloys). *A* is calculated as 0.0889. In our study, the hardness of specimens was tested under the load of 294 N. Equation (2) can be simplified as follows:(2)$$K_{IC}=1.5\sqrt{\frac{{\mathrm{HV}}_{30}}{\sum l}}$$
where HV_30_ (N·m^−2^) is the Vickers hardness of materials under the load of 30 kg, $K_{IC}$ is the fracture toughness of the material, and its unit is MPa·m^1/2^.

### 2.3. Microstructural Characterization

Sintered samples were ground and polished. A field emission SEM (SIGMA 500, Carl Zeiss AG, Jena, Germany) equipped with an energy dispersive spectrometer (EDS, SIGMA 500, Carl Zeiss AG, Jena, Germany) was used for the observation of microstructure, indention cracks and surface fractures. The X-ray diffraction (X’Pert pro, PANalytical B.V., Almelo, The Netherlands) was used to characterize the phases of the Ti(C,N)-based cermets.

## 3. Results and Discussion

### 3.1. Sintering Influences on Microstructure and Composition

The changes of the composition and microstructure of the materials were observed by scanning electron microscope (SEM). As shown in Figure 4, there are three typical core-rim structures in Ti(C,N)-based cermets.

It can be seen from Figure 5a, the relative density of the materials had not changed obviously with the sintering temperature increase. This indicated that the sintering temperature had little effect on the relative density of the materials in the range of sintering temperatures in this paper. This is illustrated in Figure 5b, the relative density of the materials increases with the holding time increase. When the holding time is 80, the compactness of the material is better than the other two holding times. The dissolution-precipitation mechanism in the sintering process can affect the compactness of materials. 

They were formed through dissolution-precipitation [1]. These cermets manifested three phases: the black core-grayish rim, black core-bright rim and bright core-grayish rim. The black cores account for a large proportion of the material. Combined with EDS results analysis, the main components of the black core are TiC and Ti(C,N) [26], and the grayish rim is composed of (Ti,W)(C,N).

As found, all these sintered samples are of a typical core-rim structure. When the temperature was at 1400 °C, the core-rim structure was completely formed, and the thickness of the rim phases was moderate. With the increase of the sintering temperature, it can be seen from Figure 6b that some particles agglomerate and the microstructure of the material becomes uneven. In the sintering process, small particles dissolve more quickly than coarse particles, so small particles are more likely to dissolve and settle on the coarse particles. When temperature increased to 1450 °C, some grains began to grow unevenly, leading to formation of large grains with sharp points (as marked by a circle), as shown in Figure 6c because of the too thick rim phase.

Figure 6 displays the microstructure of Ti(C,N)-based cermet under different temperatures of sintering.

Figure 7 displays the microstructure of Ti(C,N)-based cermet under different temperatures of sintering.

Under a holding time of 40 min, as shown in Figure 7a: The microstructure of Ti(C,N)-based cermet included the uneven black grains and large bright rim phase. In the process of liquid phase sintering, due to the short holding time, some elements such as W, Mo and Ti did not completely dissolve, thus forming a large bright rim phase. These situations lead to poor compactness of the material and affected the mechanical properties of cermet. Under a holding time of 60 min, Ti(C,N)-based cermet microstructure became more uniform. From Figure 7c, under a holding time of 80 min, the core-rim structure took shape completing, while the rim phases had moderate thickness, and the less white cores existed in the core–rim microstructures.

Figure 8 displays patterns of X-ray diffraction (XRD) of cermet generated through diversified processes of sintering.

The copper target with a wavelength of 0.154 nm was used in XRD detection. The material was scanned at a range of 20–90 degrees at a scanning speed of 1°/min. By matching peak positions and a standard XRD spectrum, the same phase composition was found in all samples although processes of sintering were different. As found, all samples had two main phases: Co/Ni solid solution phase and Ti(C,N) phase. Combined with EDS results analysis, WC and Mo2C were completely dissolved into Ti(C,N) and Co/Ni liquid phase in the sintering process [27]. Therefore, the (Ti,Mo,W)(C,N) solid and Co/Ni solution was formed in the material. Crystal structure and lattice parameters were similar to Ti(C,N). With the increase of sintering temperature, it can be seen in Figure 8 that the Ni/Co peak is shifted to a higher angle. With the increase of temperature, more hard phase particles dissolve into the binder, and more W and Ti precipitate into the edge phase. This causes the diffraction peak to shift to a high angle [28]. With the increase of the holding time, the diffraction angle of all characteristic diffraction peak for (Ti,W,Mo,Ta)(C,N) phase and Ni/Co haven’t changed significantly.

An energy dispersive spectrometer (EDS) was used to test the material composition of Ti(C,N)-based cermet (holding time was 60 min and sintering temperature was 1400 °C), as displayed in Figure 4a. The components of these three structures are shown in Table 2.

As shown in Table 2, the black cores comprised a lot of Ti and C as well as a few of Ni, Co and Mo. As confirmed already, black cores existed as remnants from non-dissolved Ti(C,N) carbide particles. Composition of (Ti,W)(C,N) in the grayish rim was caused by the dissolution and precipitation process, namely compounding of the cermet. During liquid sintering, the grayish rims took shape [29]. As shown in Table 2, the bright cores were embraced by grayish rim structures. The bright cores may be a solid solution (Ti,Mo,W)(C,N) and (Ti,W)(C,N). According to theories concerning dissolution-re-precipitation of carbides contained by Ti(C,N)-based cermets, the bright cores took shape from WC, Mo and Ti(C,N) dissolved at liquid phase during an earlier sintering stage [30].

### 3.2. Sintering Influences on Mechanical Properties

Figure 9 displays mechanical properties of Ti(C,N)-based cermets which are treated with different sintering courses.

It can be seen from Figure 9a, when the holding time is unchanged, the hardness of the Ti(C,N)-based cermet decreases as sintering temperatures increase. The highest hardness of the material is 1600 HV30 under the sintering temperature of 1400 °C. There is no obvious change for fracture toughness as sintering temperatures increase. When the sintering temperature is unchanged, holding time is prolonged from 40 min to 80 min. This is illustrated in Figure 9b, holding time in the sintering process has little influences on the hardness of Ti(C,N)-based cermet. In addition, fracture toughness of the material increases as the compactness of material increases. The highest fracture toughness of Ti(C,N)-based cermet is 9.7 MPa·m^1/2^ under a holding time of 80 min.

The results of mechanical properties analysis show that sintering temperature has a great influence on the hardness of Ti(C,N)-based ceramics. It decreases as sintering temperatures increase, but fracture toughness does not have high sensitivity towards temperatures of sintering. Besides, fracture toughness of Ti(C,N)-based cermet is greatly influenced by holding time. The fracture toughness increases as holding time increases. Consequently, Ti(C,N)-based cermet manifests excellent mechanical properties under a holding time of 80 min and the maximum sintering temperature of 1400 °C. The hardness of the Ti(C,N)-based cermet is 1600 HV30 and fracture toughness is 9.7 MPa·m^1/2^.

## 4. Conclusions

The sintering process can affect the composition, mechanical properties and microstructure of Ti(C,N)-based cermet. This paper analyzes and discusses the influence of the sintering process on the three aspects of Ti(C,N)-based cermet materials. The main conclusions of this paper are as follows:
(1)When the sintering temperature fluctuated within the scope of 1400 °C–1450 °C, some grains began to grow unevenly and large grains with sharp points appeared. The material hardness decreases as sintering temperatures increase.(2)Mainly fracture toughness of materials is influenced by holding time. As holding time increases, the microstructure tends to be uniform, while core-rim structures become complete. If the holding time is too short, the sintering process will be insufficient, leading to poor compactness of materials and large grains. As a result, the fracture toughness of materials is reduced.(3)The sintering process did not influence the phase composition but affects the phase content and crystallinity. With the increase of sintering temperature, the Ni/Co peak is shifted to a higher angle.

## Figures and Tables

**Figure 1 materials-13-03938-f001:**
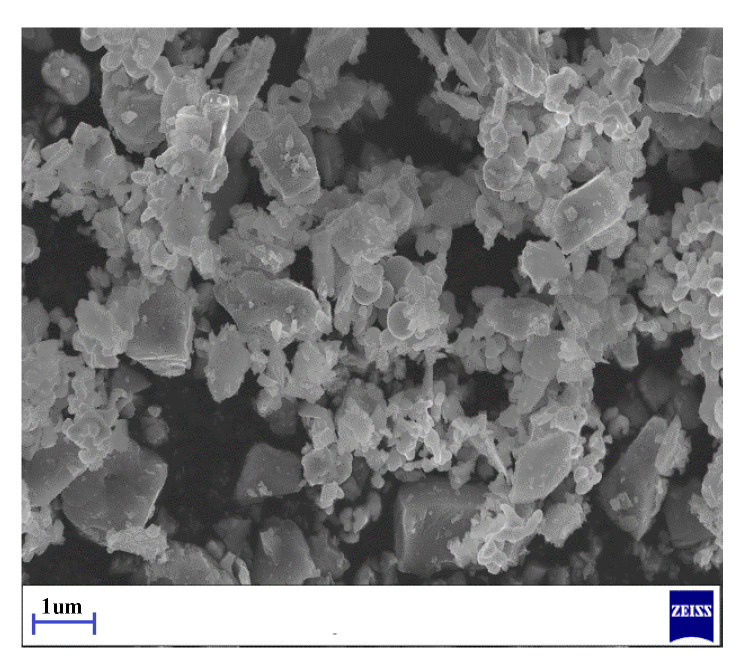
Morphology of Ti(C,N) powder.

**Figure 2 materials-13-03938-f002:**
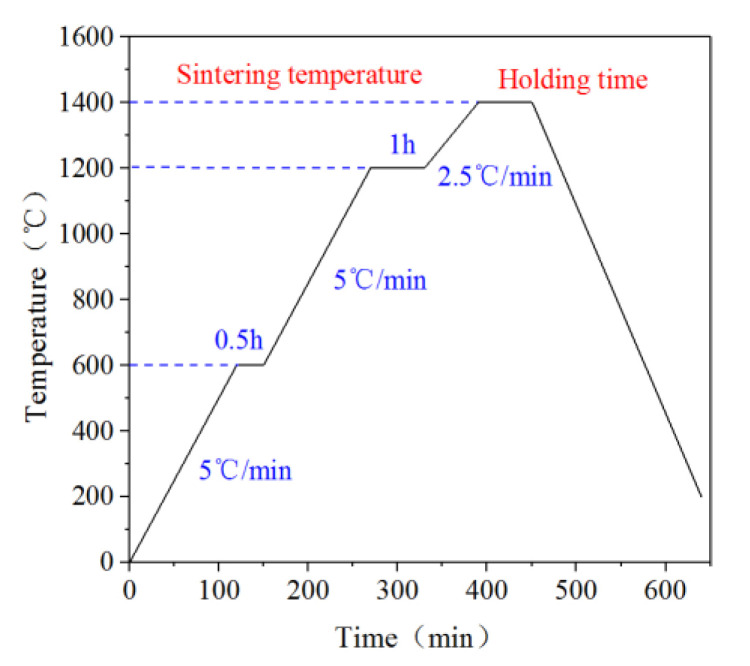
Example of sintering plan Ti(C,N)-based cermet for sintering temperature 1400 °C and sintering time 60 min.

**Figure 3 materials-13-03938-f003:**
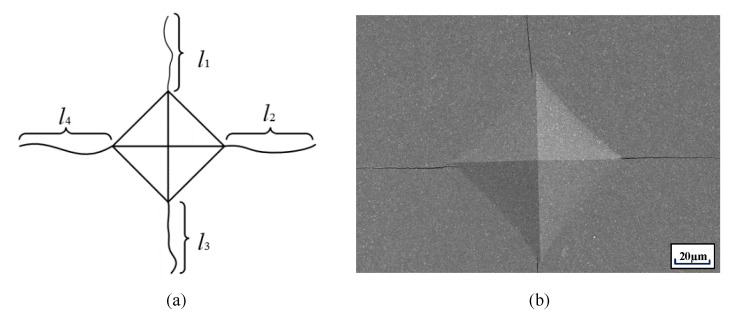
(**a**) Vickers indentation scheme with cracks enabling calculation of the critical stress intensity factor K_IC_; (**b**) the real view of the indentation with propagating cracks from the corners obtained by means of SEM.

**Figure 4 materials-13-03938-f004:**
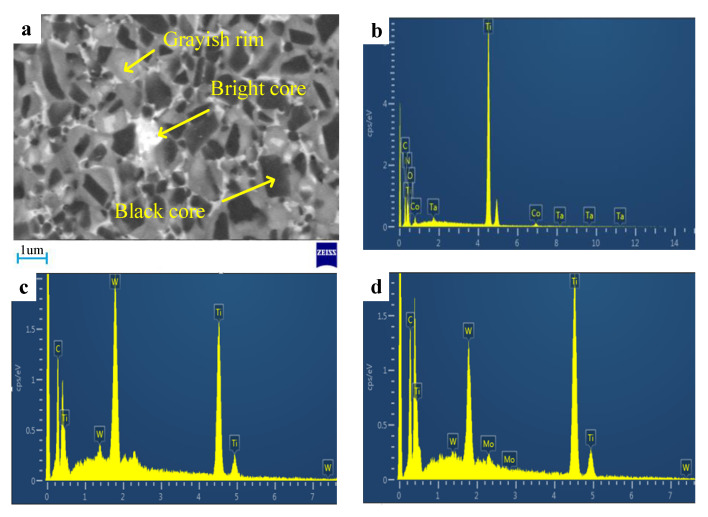
(**a**) Typical microstructure of Ti(C,N)-based cermet (sintering temperature at 1400 °C, holding time at 80 min), (**b**) black core, (**c**) grayish rim and (**d**) bright core EDS analysis of the material.

**Figure 5 materials-13-03938-f005:**
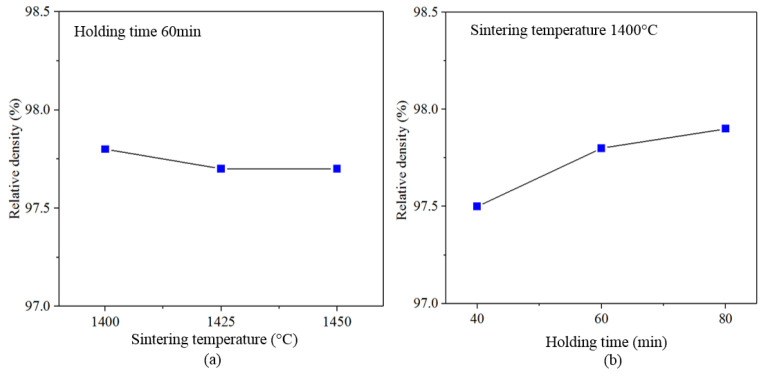
Relative density of Ti(C,N)-based cermets with different sintering processes. (**a**) different sintering temperatures when holding time is 60 min, (**b**) different holding time when sintering temperature at 1400 °C.

**Figure 6 materials-13-03938-f006:**
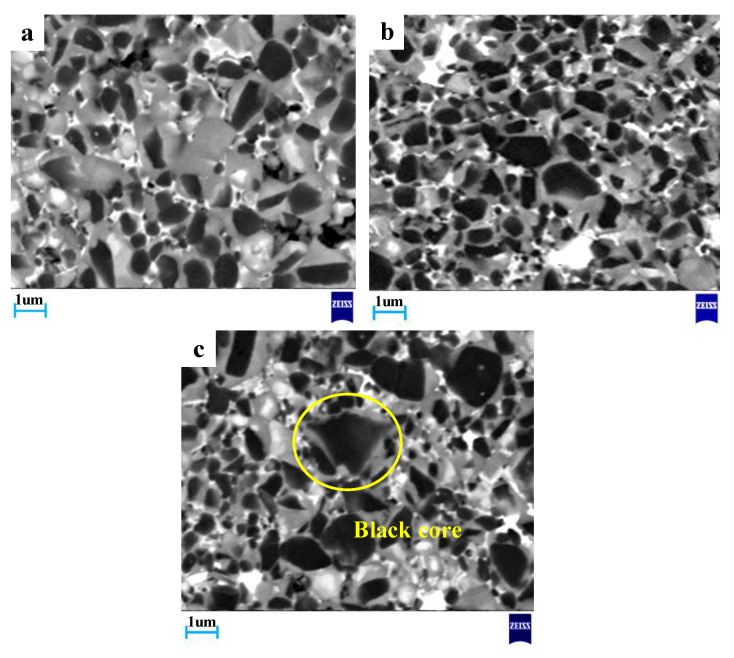
Micrographs of Ti(C,N)-based cermet sintering at different temperatures (**a**) 1400 °C; (**b**) 1425 °C; (**c**) 1450 °C.

**Figure 7 materials-13-03938-f007:**
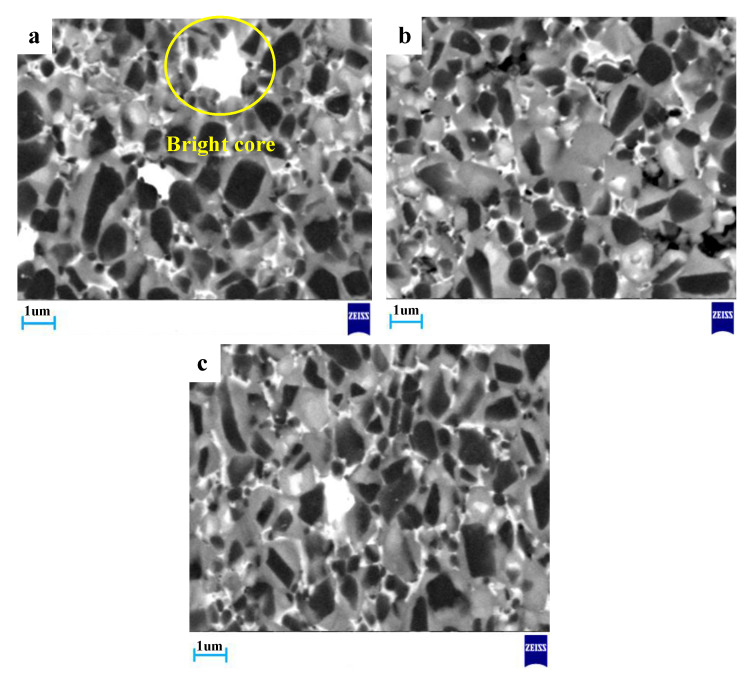
Micrographs of Ti(C,N)-based cermet sintering at different holding times (**a**) 40 min; (**b**) 60 min; (**c**) 80 min.

**Figure 8 materials-13-03938-f008:**
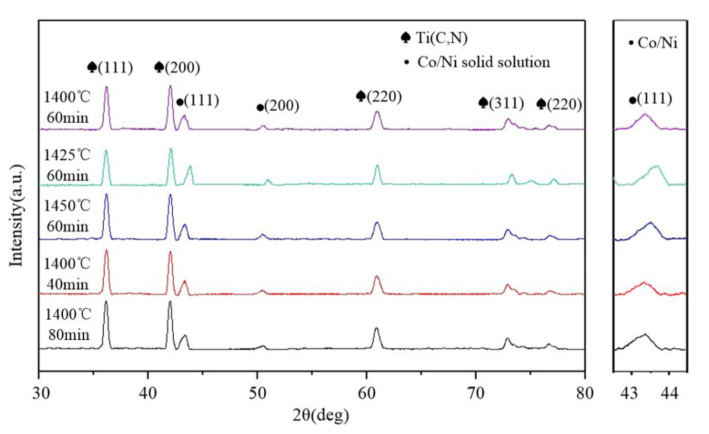
X-ray diffraction patterns of (Ti,M)(C,N)-based cermets.

**Figure 9 materials-13-03938-f009:**
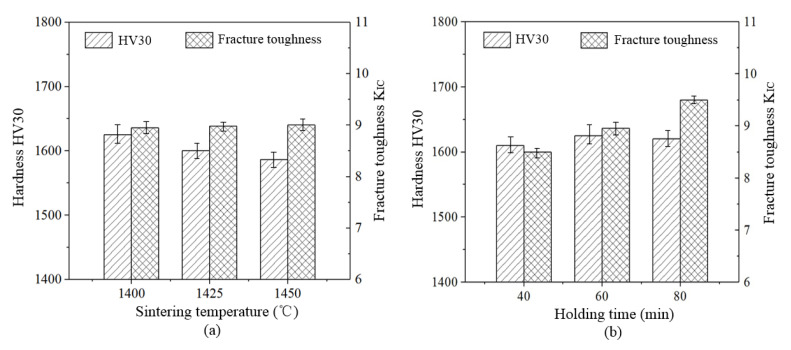
Mechanical properties of Ti(C,N)-based cermets sintered at different sintering processes: (**a**) sintering temperature effect, (**b**) holding time effect at 1400 °C.

**Table 1 materials-13-03938-t001:** Properties of the Ti(C,N)-based cermet raw powders.

Powders	Grain Size (d_50_/µm)	Purity (wt.%)	Impurity (wt.%)			Manufacturer
Ti(C_0.5_,N_0.5_)	0.5	99.6	O (<0.08)	Na (0.004)	Al (0.002)	Beijing Xiongrongyuan Technology Co., Ltd. (Beijing, China)
WC	0.4	99.6	O (<0.08)	N (0.002)	Al (0.002)	Xiamen Golden Egret Special Alloy Co., Ltd. (Xiamen, China)
TaC	0.5	99.8	O (<0.20)	N (0.002)	K (0.002)	Shanghai Chaowei Nanotechnology Co., Ltd. (Shanghai, China)
MoC	0.4	99.9	C (0.01)	Fe (0.005)	Si (0.003)	
Ni	0.8	99.9	S (0.004)	Fe (0.004)	O (<0.004)	
Co	2.2	99.0	O (0.23)	Fe (0.019)	C (0.09)	

**Table 2 materials-13-03938-t002:** Composition of cermets during EDS analysis.

Element (at.%)	Ti	C	W	Mo	Ta	N	Co	Ni
Black core	50.08	9.87	15.24	2.36	2.24	5.62	6.89	7.69
Grayish rim	59.05	11.15	29.80	0.2	0.	0	0	0
Bright core	62.6	10.98	15.47	1.51	0.2	9.43	0	0

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
