# Peer review of "Influence of Sintering Process on Microstructure and Mechanical Properties of Ti(C,N)-Based Cermet"

_materials, 2020, doi:10.3390/ma13183938_

Round 1
Reviewer 1 Report
Novelty
Present paper lacks novelty since analogous investigations about the effect of sintering properties on the structure and mechanical properties of similar TiC(N) based cermets have been published before. In this work the TiCN-CoNi cermets are alloyed with WC, MoC and TaC, but almost no discussion regarding the effect of alloying compounds is given.
- Experimental details and processes
The manufacturer of the powders should be given. In table 1 only countries are listed and it is not correct to label them as “manufacturer”. In addition, correct grain size units should be used (µm, not um).
Please provide the description of composition of starting powder! What is metal and carbide ratio? What is starting composition of metal phase?
LINE 64: More correct term is “morphology” instead of “microstructure”.
LINES 68-69: First it is said that milling speed was 200 r/min and afterwards 160 rpm. Which speed was used? Avoid using different spelling for same units.
LINES 71-73: Difficult to understand the description of holding times.
LINE 84: Vickers tester is same model as milling device??
Equipment: You have presented some information regarding used equipment (ball mill, OM, hardness tester). However, no information about sintering furnace, SEM, EDS and XRD devices is given.
Please describe the sintering parameters more thoroughly. What was the vacuum level in the furnace during sintering? During vacuum sintering the TiCN phase can start to decompose (loss of nitrogen) and it is dependent on the vacuum level. What is the material of sintering chamber and heating elements?
- Results and discussion
As a reader, I would suggest that microstructure is discussed before the mechanical properties. The structure is directly the result of the preparation methods and mechanical properties are defined by the microstructure. In this paper you present the mechanical properties, but discussion (why hardness decreases with increase in temperature and toughness increases with holding time) is given in later chapter about microstructure. Therefore, it makes sense to present mechanical properties after discussion about microstructure.
Based on XRD data you claim that content of Co and Ni is affected by sintering parameters. This claim is misleading. While the dissolution of carbide forming elements can increase and therefore increase the intensity of XRD peaks of binder phase, the content of Co and Ni remains the same. In addition, the crystal size affects the intensity of respective XRD peaks. Therefore, the higher peak intensity at higher temperature may not be the result of changes in chemical composition of metal phase at all.
What is the cause of XRD peaks shift (Co/Ni phase)?
Since no grain size measurements were presented it is incorrect to make conclusions regarding sintering parameters and changes in grain size, especially using only SEM images with relatively high magnification (10000x).
Information regarding porosity should be added. How porosity of investigated TiCN-based cermets is influenced by sintering parameters?
Grammar
Extensive review of grammar is needed. Past and present tenses are used together in same paragraphs (for example LINES 70 and 71). Sentence construction must be improved as-well. I would recommend proof-reading by native english speaker.
Reviewer 2 Report
The reviewed work concerns the manufacturing process of Ti (C, N) -based cermet with a description of technological parameters and their influence on structure changes and the obtained mechanical properties. It is written correctly and clearly, but in the light of the work published so far on this topic, it does not bring much new in terms of Ti (C, N) -based cermet. In addition, I have noticed some shortcomings or inaccuracies which are presented below:
Line 33-35: Why did the authors omit the SPS method?
Line 66: Only one picture of the microstructure is shown, why is the plural in the signature? In addition, it is known that this is a SEM image. What did the authors want to show on it, morphology or some other characteristic features of the powder? The signature under the photo should bring something, so as not to be just a trivial filler.
Line 67-69: What was the milling rate of 160rpm or 200rpm?
Fig. 2. The caption under this figure should read: Example sintering plan Ti (C, N) -based cermet for sintering temperature 1400 C and sintering time 60min.
Fig. 3. I propose to change the caption under figure 3 to: Vickers indentation scheme with cracks enabling calculation of the critical stress intensity factor KIc; (b) the real view of the indentation with propagating cracks from the corners obtained by means of SEM
Formula (1): On what basis was the formula used? What is the nature of crack propagation? Is it Palmqvist, Intermediate or Median cracking mode?
Fig.4 (b): holding time effect at 1400C
Line 119: What kind of XRD lamp was used, what was the scanning step?
Fig. 5.: The quality of the diffractograms presented is unsatisfactory. The applied smoothing caused that the shape of the obtained reflections is unnatural. Take a look at the reflections (111) of the Co/Ni binder, or the reflections (200) for a temperature of 1450C / 60 min. Reflex (310) Ti (C, N) is marked incorrectly. What's going on with reflex (220) Co/Ni? Please comment, why the diffraction patterns were not presented in the same form as in the paper [26]
Line 138: The XRD measurement shows only two phases. Please comment and compare the results of XRD and EDS. Compare your results with work https://doi.org/10.1016/j.vacuum.2019.108983
Table 2.: The EDS analysis cannot be applied to light elements such as C and N. In the case of analysis with one light element, the software determines the share of this element as a 100% complement. In this case, when quantifying two light elements, the obtained result is completely unreliable. A WDS detector should be used for C and N analysis.
Line 143-153: Compare with the work of ZHOU Shu-zhu, et al / Trans. Nonferrous Met. Soc. China 19 (2009)
Line 157-165: These are more speculations than analysis results. There is no evidence submitted for such a mechanism.
Line 166: The theory of Hall-Petch talks about changing the yield strength as a function of grain size. Generalizing it for bending strength is an abuse
Line 167-168: On what grounds was this statement made? Was the grain size analyzed? Or maybe the decrease in hardness is the effect of changing the share of individual phases?
Line 175-177: There is no evidence for such a statement. He can perform microanalysis of the chemical composition in specific areas for specific temperature and time states and only carry out analyzes of changes occurring in the structure of the material.
Line 185-186: Where is this visible? On the attached XRD patterns the strongest reflections for Co/Ni occur at 1425C / 60min.
Line 200-201: Where is this visible? On the attached XRD patterns the strongest reflections for Co/Ni occur at 1425C / 60min.
Reviewer 3 Report
The K. Ji et al. studied how the sintering time and temperature influence the microstructure and mechanical properties of Ti(C,N)-based cermets. The experimental details and related interpretations are well presented. The results can be useful for understanding and improving the properties of cermet materials. I would recommend publication after the following comments are considered.
1. Why the authors use Ti(C, N) powders contain Co, Ni, Mo, W, instead of pure Ti(C, N) as a model system to study the influence of sintering time and temperature? Do the authors have control experiments with Ti(C.N)?
2. Since the alloying with other metals such as W, Mo can greatly change the sintering activity and properties of Ti(C, N). Can the authors discuss how this effect influence on their results? And the authors should give the actual stoichiometrical compositions in the text.
Reviewer 4 Report
Row 73
This paper used five .. .Please correct this sequence
Row 83-84
Please correct this sentence.
Row 97
The equation needs to be corrected. Summe of l is the mean value of radial crack
Round 2
Reviewer 2 Report
Dear Authors,
Thank you for the explanations sent, most of them satisfy me. However, I still have a few doubts:
Regarding the KIc measurements. The authors cite the work [26] by Shetty, D.K .; Wright, I.G.; Mincer, P.N .; Clauer, A.H. Indentation fracture of WC-Co cermets. J MATER SCI 1985, 20, 1873–1882. However, there is still no evidence that the cracks formed are Palmqvist in nature. The fact that in the cited work they had such a form does not prove that it is also the same in this work. I suggest polishing the sample in the area of the indentation, which will clearly show the nature of the cracks. How was crack resistance parameter, W determined? Was the Shetta et al. model used or a modified model (see the work of Y. K. Song, R. A. Varin, Indentation microcracking and toughness of newly discovered ternary intermetallic phases in the Ni-Si-Mg system, Intermetallics 6 (1998) 379-393)? If a modified model was used, how was the value of the P0 force causing crack propagation determined? Please also present the calculations of how the final form of formula (1) was determined. In addition, according to ISO 6507-1: 2018, the hardness value when measured with a force of 30 kgf should be recorded as HV30.
Please delete line 174.
How was the theoretical and real density determined? What method and how many measurements? Why is the scale on the Y axis in Figures 8a and 8b different? For what time is Fig.8a, and for what temperature Fig.8b?
Round 3
Reviewer 2 Report
Dear Authors,
Thank you for the explanations sent, the posted photo explains that they are Palmqvist cracks. However, please pay attention to the correctness of the presented calculations. Which value of β is correct 0.00889 or as given in formula (4) 0.0889. For β = 0.0889 formula (5) will not have this form.
Author Response
请参阅附件
